# The Cost-Effectiveness of the SMART Work & Life Intervention for Reducing Sitting Time

**DOI:** 10.3390/ijerph192214861

**Published:** 2022-11-11

**Authors:** Edward Cox, Simon Walker, Charlotte L. Edwardson, Stuart J. H. Biddle, Alexandra M. Clarke-Cornwell, Stacy A. Clemes, Melanie J. Davies, David W. Dunstan, Helen Eborall, Malcolm H. Granat, Laura J. Gray, Genevieve N. Healy, Benjamin D. Maylor, Fehmidah Munir, Thomas Yates, Gerry Richardson

**Affiliations:** 1Centre for Health Economics, University of York, York YO10 5DD, UK; 2Diabetes Research Centre, University of Leicester, Leicester General Hospital, Leicester, LE5 4PW, UK; 3NIHR Leicester Biomedical Research Centre, Leicester LE5 4PW, UK; 4Centre for Health Research, University of Southern Queensland, Springfield, QLD 4300, Australia; 5School of Health & Society, University of Salford, Salford M6 6PU, UK; 6School of Sport, Exercise and Health Sciences, Loughborough University, Loughborough LE11 3TU, UK; 7Leicester Diabetes Centre, University Hospitals of Leicester, Leicester LE5 4PW, UK; 8Baker Heart and Diabetes Institute, Melbourne, VIC 3004, Australia; 9Baker-Deakin Department Lifestyle and Diabetes, Deakin University, Melbourne, VIC 3004, Australia; 10Department of Health Sciences, University of Leicester, Leicester LE1 7RH, UK; 11Deanery of Molecular, Genetic and Population Health Sciences, The University of Edinburgh, Edinburgh EH16 4UX, UK; 12School of Human Movement and Nutrition Sciences, The University of Queensland, Brisbane, QLD 4067, Australia

**Keywords:** cost-effectiveness, SMART, sedentary behaviour, healthy habits, standing desks

## Abstract

Sedentary behaviours continue to increase and are associated with heightened risks of morbidity and mortality. We assessed the cost-effectiveness of SMART Work & Life (SWAL), an intervention designed to reduce sitting time inside and outside of work, both with (SWAL-desk) and without (SWAL-only) a height-adjustable workstation compared to usual practice (control) for UK office workers. Health outcomes were assessed in quality-adjusted life-years (QALY) and costs in pound sterling (2019–2020). Discounted costs and QALYs were estimated using regression methods with multiply imputed data from the SMART Work & Life trial. Absenteeism, productivity and wellbeing measures were also evaluated. The average cost of SWAL-desk was £228.31 and SWAL-only £80.59 per office worker. Within the trial, SWAL-only was more effective and costly compared to control (incremental cost-effectiveness ratio (ICER): £12,091 per QALY) while SWAL-desk was dominated (least effective and most costly). However, over a lifetime horizon, both SWAL-only and SWAL-desk were more effective and more costly than control. Comparing SWAL-only to control generated an ICER of £4985 per QALY. SWAL-desk was more effective and costly than SWAL-only, generating an ICER of £13,378 per QALY. Findings were sensitive to various worker, intervention, and extrapolation-related factors. Based on a lifetime horizon, SWAL interventions appear cost-effective for office-workers conditional on worker characteristics, intervention cost and longer-term maintenance in sitting time reductions.

## 1. Introduction

Sedentary behaviours continue to increase and are associated with heightened risks of several chronic diseases and all-cause mortality [1]. In the United Kingdom (UK), sedentary behaviours are known to contribute to over £700 million in National Health Service (NHS) costs and 69,276 deaths in 2016 [2]. The COVID-19 pandemic has further exacerbated sedentary behaviours and their consequences for public health [3,4], prompting policy interest in public health strategies that promote safe physical activity and reduce sedentary behaviours [5,6]. 

In line with the global context, UK office workers are a highly sedentary population, spending approximately 70–85% of their time at work sitting [7,8,9]. With over half of the UK working population economically active and predominately working in sedentary or light physical activity occupations, workplaces are an ideal setting for interventions designed to reduce daily sitting. Acute experimental studies have shown strategies designed to promote frequent bouts of light-intensity movement can improve markers of cardiometabolic and musculoskeletal health [10,11,12,13,14,15,16], while meta-analytic studies have found the relationship between sedentary time and adverse outcomes are most pronounced at the highest levels of inactivity [17,18,19]. Both public- and private-sector organisations need to identify and manage the wider risks of sedentary behaviour to people’s health. Embracing a preventive health model stands to improve office-workers’ physiological and psychological health, alleviate public healthcare requirements, and assist in the development of a resilient and productive workforce [20,21]. 

The SMART Work & Life programme is a multicomponent intervention, designed to reduce ambulatory office workers’ sitting time inside and outside work [22]. The intervention includes organisational, environmental, and group/individual level behaviour change strategies, delivered by workplace champions. The programme’s effectiveness was tested within a cluster randomised controlled trial, which demonstrated that SMART Work & Life successfully reduced daily sitting time compared to a control group. This economic analysis aims to consider the cost-effectiveness of the SMART Work & Life intervention, delivered with and without a height-adjustable workstation, using evidence from the SMART Work & Life cluster randomised controlled trial. By estimating the health benefits and costs associated with SMART Work & Life, we can aid the development of this and other public health initiatives while ensuring resources are allocated only to those interventions which maximise population health [23,24].

## 2. Materials and Methods

### 2.1. Overview

The cost-effectiveness of the SMART Work & Life (SWAL) intervention with (SWAL-desk) and without (SWAL-only) a height-adjustable workstation was evaluated in two parts: (i) a within-trial analysis considering costs and outcomes estimated over the trial period (12 months); and (ii) a lifetime horizon decision analytic modelling analysis incorporating longer-term mortality benefits from reductions in sedentary behaviour over a person’s lifetime.

Intervention and healthcare costs were measured in UK pounds sterling (2019–2020) from a public sector perspective [25]. Office-worker outcomes included quality adjusted life-years (QALY), absenteeism days, measures of productivity, psychological health, job satisfaction, and work engagement [7,26]. In line with UK guidelines, costs and QALYs were discounted at 3.5% per annum [25]. Within-trial costs and QALYs were estimated using regression methods to control for participant co-variables. Long-term outcomes were modelled using contemporary estimates of the dose–response relationship between sedentary time and all-cause mortality [18,19]. Cost-effectiveness results are presented as incremental cost-effectiveness ratios and incremental net-health benefits at cost-effectiveness thresholds relevant to UK decision-making (£15,000, £20,000 and £30,000 per QALY) [24]. Missing cost, QALY and absenteeism data were imputed [27,28]. Probabilistic sensitivity analysis was used to estimate decision uncertainty. Deterministic scenario, sensitivity, and threshold analyses further explored the impacts surrounding key model assumptions.

### 2.2. SMART Work & Life Trial

The SMART Work & Life trial was a three-arm cluster randomised controlled trial (n = 756) that evaluated the effectiveness of SWAL-desk (n = 240) and SWAL-only (n = 249) at reducing daily sitting time in office workers compared to a control (usual practice) (n = 267). SWAL includes organisational, environmental (e.g., relocating waste bins, printers), and group/individual (e.g., education, action planning, goal setting, addressing barriers, group coaching, sitting less challenges, self-monitoring) level behaviour change strategies, facilitated by workplace champions. Office groups within local councils across three areas of England were randomised to one of three arms: SWAL-desk, SWAL-only, or control. Randomisation was stratified by council area (Leicester, Liverpool, and Greater Manchester) and cluster size (<10 and ≥10 participants). Trial participants consisted of consenting, English speaking, non-pregnant and mobile (able to walk unassisted) adult office workers (≥60% full time equivalent) nested within shared office spaces [26]. Follow-up measurements were taken at 3 and 12 months with activity data recorded using a thigh-worn accelerometer-based device. Further details about SWAL and the trial are available elsewhere [29].

### 2.3. Resource Use and Costs

Costs for each trial participant were categorised into (i) intervention-related; (ii) health-related; and (iii) absenteeism-related. Intervention-related costs were calculated on an intention-to-treat (ITT) basis and comprised: training workplace champions and general staff (including staff time costs); managers correspondence time; office motivational materials; and the procurement and installation of height-adjustable workstations. Health-related costs were assessed from a public-perspective and equated to the number of self-reported resources consumed during the trial multiplied by their respective unit costs. Linear interpolation was used to populate resource use data not recorded between 3-month and 9-month follow-up. Unit costs were obtained from a variety of published UK sources (Appendix A) and were inflated to UK pounds sterling 2019–2020 where necessary [30,31]. Absenteeism-related costs were calculated using the median daily UK salary (2019) [32]. 

### 2.4. Outcomes

The primary health outcome for office-workers was expressed by QALYs, a generic measure of health where one QALY represents a year in perfect health [24]. Within-trial health-related quality of life (HRQoL) was measured using EQ-5D-5L responses, a descriptive instrument defined by five health dimensions (mobility; self-care; usual activities; pain/discomfort and anxiety/depression) assessed over five severity levels (no problems, slight problems, moderate problems, severe problems and extreme problems) [33]. As recommended in NICE guidelines, EQ-5D-5L responses were mapped onto those applicable to the UK EQ-5D-3L value set to generate HRQoL weights [25,34]. HRQoL weights from the UK EQ-5D-5L value set were considered as a scenario. Observed within-trial QALYs were calculated using an area under the curve approach with linear interpolation between time points. QALYs beyond the trial were calculated by the decision analytic model. Measures of stress, wellbeing, absenteeism, and work performance, satisfaction and engagement were considered descriptively.

### 2.5. Analysis

#### 2.5.1. Within-Trial Analysis

Within-trial costs and QALYs were estimated from derived observations (see Section 2.2 and Section 2.3) for each treatment using generalised linear regression models that controlled for age, gender, ethnicity (White vs. Other), body mass index (BMI), site area (Leicester; Salford; Liverpool) and cluster size (small < 10; large ≥ 10) [35]. QALY and absenteeism regression analyses also controlled for baseline EQ-5D scores and absenteeism days, respectively [36]. Costing and absenteeism regressions used a log-link transformation and gamma family form to account for the bounded right-skewed nature of the dependant variable [37]. QALY regressions applied ordinary least squares. Multi-level regression models that considered residual components at the site-level hierarchy were considered in scenario analyses [38].

#### 2.5.2. Decision Analytic Model and Long-Term Cost-Effectiveness

Outcomes beyond the trial were estimated using a two-state alive-dead Markov-model with costs and QALYs discounted at 3.5% per annum, and 1.5% in a scenario [25,39]. In the alive-state, individuals experience HRQoL equal to age-adjusted English population norms and accrue no costs [40]. A scenario considered age-specific NHS costs for the alive state [41]. For control, transitions to the death-state were assumed to follow age- and sex-adjusted English general population mortality rates [42]. Transitions were adjusted by common all-cause mortality hazard ratios (i.e., those irrespective of participant characteristics) associated with each SWAL-interventions’ reduction in sedentary time, as reported in Ekelund et al.’s accelerometer-measured meta-analysis (see Appendix A) [18]. Hazard ratios from Patterson et al.’s (2018) broader meta-analysis (including non-objective sedentary measurements) were considered as a scenario [19]. In line with previous economic evaluations, treatment-effects on sedentary time were assumed to exponentially decline at a 50% decay rate per annum (i.e., every year the reduction in sedentary time associated with SWAL interventions is halved) [43,44,45,46]. Alternative exponential decay rates and a linear decay rate of 20% per annum were explored in scenario and sensitivity analyses [47]. 

#### 2.5.3. Statistical Methods

Missing trial cost, QALY, and absenteeism data were imputed using a multi-level model to account for heterogeneity between sites and controlled for all the covariates considered in the within-trial regression models. [27]. Estimates generated across 20 imputed datasets were combined using Rubin’s Rules [48,49]. Imputed QALYs were bounded at one and costs and absenteeism days were bounded to positive values. Decision uncertainty was propagated using Monte Carlo simulation assuming normality in sedentary times and treatment effects and multivariate normality of regression coefficients [50]. Uncertainty in mean cost, QALY, and cost-effectiveness estimates was reported via 95% credible intervals and probabilities of being most costly, effective, and cost-effective. 

#### 2.5.4. Economic Analysis

Cost-effectiveness was assessed over the trial period (12 months) and over a lifetime horizon using incremental cost-effectiveness ratios (ICERs) and incremental net health benefits (INHBs) [24]. ICERs represent the incremental cost per additional QALY of a strategy compared with the next best alternative; INHB the SWAL-related health gain relative to control less the health opportunity cost from additional expenditure as defined by the cost-effectiveness threshold (i.e., the health forgone elsewhere from resources not being available for other health generating purposes). INHB and the probability of being cost-effective are presented at three relevant UK threshold values: £15,000, £20,000 and £30,000 per QALY [25,51,52]. ICERs below the threshold are deemed cost-effective (compared to the next best relevant comparator); a positive INHB infers cost-effectiveness compared to control. 

Additional scenario analyses included alternate worker age profiles, removing estimated within-trial public sector costs and QALY differentials between arms, and a broader multi-sectoral perspective including private absenteeism costs. When including private absenteeism costs, two scenarios were considered; one where they were weighted equally with public sector costs and another where private sector costs are valued at a lower rate based on evidence of the higher opportunity costs associated with public resources (private costs weighted four times lower) [51,52,53,54]. Sensitivity analyses considered how changes in age, treatment decay rates, treatment costs, and treatment-associated reductions in sitting times impacted the INHB of each SWAL intervention compared to control.

## 3. Results

### 3.1. Missing Data

Over the course of the trial, the degrees of data completion for the resource use, EQ-5D-5L and absenteeism follow-up data ranged between 61.8–83.5%, 69.0–93.0% and 65.6–87.4%, respectively. Participant characteristics are reported elsewhere [26]. 

### 3.2. Outcomes

SWAL-desk and SWAL-only were associated with 63.7- and 22.2 min per day less sitting time compared to the control group at 12-month follow-up, respectively. Imputed within-trial EQ-5D-5L scores, mapped EQ-5D-3L values and associated QALY outcomes were broadly comparable across the trial arms (Appendix A). When controlling for participant covariates, QALYs were highest for SWAL-only followed by the control arm and SWAL-desk (Appendix A). The number of employee-reported sick days were similar across the comparators, albeit with temporal changes more favourable for control relative to SWAL-related interventions. Modest improvements in stress and wellbeing scores were observed, whilst no marked differences in job satisfaction, job performance, and measures of work engagement were found (Appendix A). Controlling for baseline values and participant covariates, absenteeism days were highest for SWAL-desk followed by SWAL-only and control (Appendix A). At model baseline (12-month follow-up), reductions in sitting time for SWAL-desk and SWAL-only translated into 23.8% and 10.5% reductions in the relative-risk of all-cause mortality compared to control, respectively.

### 3.3. Resource Use and Costs

The average intervention cost per ITT individual was £228.31 and £80.59 for SWAL-desk and SWAL-only, respectively (Appendix A). Healthcare resource use was broadly balanced between trial arms (Appendix A). Imputed within-trial healthcare costs were lowest for SWAL-desk (£541.24) compared to SWAL-only (£672.58) and control (£658.78). When controlling for participant covariates, healthcare costs were highest in the control arm, followed by SWAL-only and SWAL-desk (Appendix A). Imputed absenteeism costs were highest for SWAL-desk (£211.58) followed by SWAL-only (£184.48) and control (£139.97). This cost ordering was maintained when controlling for baseline values and participant covariates (Appendix A). 

### 3.4. Economic Analysis

#### 3.4.1. Cost-Effectiveness

Inputs used in the cost-effectiveness analysis are reported in Appendix A. Table 1 displays the adjusted within-trial and lifetime horizon cost, QALY and cost-effectiveness findings. The within-trial analysis found SWAL-desk to be dominated by both SWAL only and control (i.e., SWAL-desk being the most costly and the least effective alternative). SWAL-only was more costly and effective than control with an ICER of £12,091 per QALY; an INHB range of 0.0011–0.0025; and a probability of being cost-effective ranging between 42.3–43.5% (for a £15,000-£30,000 per QALY threshold range). The lifetime horizon analysis found SWAL-desk to be the most costly and effective (due to reductions in sedentary time); control the least costly and least effective; and SWAL-only the second-most costly and effective. SWAL-only generated an ICER of £4985 per QALY; an INHB range 0.007–0.008; and a probability of being cost-effective range 36.4–38.8% (for a £15,000–£30,000 per QALY threshold range). SWAL-desk was cost-effective compared to SWAL-only at the threshold values considered (ICER: £13,378 per QALY; INHB range 0.007–0.011; probability of being cost-effective range: 44.8–52.7%). The considerable overlap in the credible intervals for costs and QALY between alternatives suggests a significant level of uncertainty. 

#### 3.4.2. Scenario Analyses

Table 2 presents the results from the scenario analyses. The cost-effectiveness of SWAL interventions improved with lower treatment decay rates, suggesting that the longer the maintenance of the sitting reduction, the greater the benefits. For older individuals, the cost-effectiveness of the intervention improved as a result of greater absolute reductions in mortality. Using alternative estimates of impacts of sitting time on all-cause mortality resulted in SWAL-desk being dominated. Other scenarios had minimal impact on our cost-effectiveness findings. 

#### 3.4.3. Sensitivity and Threshold Analyses

Findings were highly sensitive to the treatment efficacy decay rate and worker age (Figure 1). SWAL-only had positive INHB across all decay rates (0–90%), study ages (30–70 years-old) and for any level of sitting-time reduction (due to modest within-trial gains in HRQoL). SWAL-only remained cost-effective at programme costs of £191 (£349) per employee at a £15,000 (£30,000) per QALY threshold. SWAL-desk had positive INHB (vs. control) at decay rates ≤ 88.4% (≤100%), ≥33 years of age, and for ≥29 min (≥15 min) reductions in sitting time at a £15,000 (£30,000) per QALY threshold. The INHB was highest for SWAL-desk at decay rates below 49.8% (61.0%), ages ≥ 44 (≥40) years and incremental (desk-related) costs relative to SWAL-only of £146 (£202) at a £15,000 (£30,000) per QALY threshold. Sensitivity analyses surrounding treatment efficacy decay rates and worker ages at thresholds £20,000 and £30,000 per QALY are reported in Appendix A. 

## 4. Discussion

### 4.1. Key Findings

Results of this study suggest both SWAL interventions are potentially cost-effective, with SWAL-only appearing cost-effective over the trial period and over a lifetime horizon compared to control, whereas SWAL-desk only appeared cost-effective using a lifetime horizon (versus control and SWAL-only). The cost-effectiveness of SWAL-only was predicated on immediate within-trial gains in HRQoL and longer-term benefits from reductions in sitting time. The overall lifetime cost-effectiveness of SWAL-desk resulted from moderate within-trial health-related cost savings and longer-term benefits from reductions in sitting time. The cost-effectiveness of both SWAL-interventions was, however, sensitive to treatment-associated within-trial cost and HRQoL impacts, participant age, maintenance of the sitting reduction, the mortality risks associated with sedentary behaviour, and intervention costs. Compared to control, SWAL-desk was associated with 14.07 incremental discounted QALYs per 1000 employees enrolled, at a cost of £105,542 from a public perspective. Compared to SWAL-only, SWAL-desk could be considered cost-effective from the public perspective provided desk-related costs fall below £202 and those additional reductions in sedentary time associated with its application are adequately maintained (≥39% per annum).

### 4.2. Previous Findings

Munir et al.’s cost–benefit analysis of SMArT Work, an earlier iteration of the SWAL programme, found the intervention improved worker self-perceived productivity and was deemed cost-effective from an organisational perspective [21]. The SWAL interventions however, found no meaningful changes in worker productivity. Gao et al.’s economic evaluation of ‘Stand up Victoria’, another multi-component intervention designed to reduce sitting time at work in desk-based staff, found the intervention was associated with reduced workplace sitting time, no significant benefits in absenteeism or self-reported HRQoL, increased costs, and improved long-term health outcomes [47]. ‘Stand up Victoria’ was deemed cost-effective over a lifetime horizon in the Australian context although results were highly sensitive to the maintenance of the reduction in sitting time. The intervention costs, study findings and conclusions are broadly comparable with those found for SWAL-desk. Michaud et al.’s within-trial cost-effectiveness analysis of ‘Stand and Move at Work’ with (STAND+) versus without (MOVE+) a standing desk, found STAND+ achieved significant reductions in sitting times (47.7 min daily) and improvements in self-reported HRQoL (0.013 EQ-5D score) at a comparable non-annuitized cost ($375) to SWAL-desk (£229) [55]. STAND+ was deemed cost-effective in the US context, despite not being associated with improvements in light-intensity physical activity, productivity or absenteeism. 

In line with our study findings, there is limited evidence that SWAL or similar work-place interventions improve employee absenteeism. Nevertheless, work-place interventions generally appear cost-effective, either as a result of health gains from reductions in sedentary time [47,56] and/or improvements in worker HRQoL [55], or via improvements in workplace productivity [21] or presenteeism [57]. Our study reinforced cost-effectiveness via health gains from reductions in sitting time. As reported in Nguyen et al.’s recent systematic literature review, cost-effectiveness evidence of sedentary behaviour reduction interventions in workplaces is limited, but appears broadly consistent [58]. 

### 4.3. Strengths and Limitations

Strengths of this study include the application of accelerometer measured treatment-specific sedentary times from a randomised controlled trial; a relatively large sample of diverse UK office workers from which costs were compiled using relevant UK costing sources; HRQoL from a validated instrument; and sedentary-associated all-cause mortality risks aligned to contemporary, robust and objective estimates [18]. In addition, study methodology was consistent with NICE methodological guidance, and the robustness of study results were tested using a wide range of scenario, sensitivity and threshold analyses [25]. All these factors support the validity of the findings observed in this study.

Limitations include the acknowledgement that the estimated within-trial treatment effects on costs and outcomes were highly uncertain and the trial was not powered to detect for such differences. Furthermore, common average treatment-effects and sedentary-associated all-cause mortality risks may overgeneralise the impacts treatments have on the consequences of sedentary behaviour across office-workers. We also applied a narrow scope of the longer-term health benefits associated with changes in sedentary behaviour were captured (i.e., all-cause mortality only) and the base case decay rate in treatment efficacy was unknown and based on those chosen by analysts in previous studies [43,44,45,46] rather than empirical data. Other limitations include: the constant sedentary times assigned to control, and those which SWAL-interventions converge to, ignore potential dynamics over-time; intervention costs did not consider potential costs beyond the trial (e.g., equipment maintenance, training and staff work time); and the trial’s setting and exclusion criteria could have limited the generalisability of our findings to other applicable contexts. Thus, conclusions regarding the cost-effectiveness of the SWAL programmes must be interpreted with caution.

### 4.4. Policy Implications

Compared to usual practice, our analysis forecasts a 9.85 and 14.07 incremental discounted QALY gain at a public cost of £49,130 and £105,542 for every 1000 office workers enrolled onto SWAL-only and SWAL-desk, respectively. These findings should help to inform resource allocation priorities, with potential implications to the health of office-workers. The extent to which SWAL interventions pose a cost-effective investment in public health is, however, uncertain and may differ in practice from those reported in our analysis for several reasons. First, the expected cost of delivering SWAL-only or SWAL-desk is subject to several market and contextual factors. Changes in the prices of constituent elements (e.g., desks, materials, training), economies of scale, local office factors (e.g., shared desks, staff time costs, remote working, office culture, work demands) and broader programme alterations (e.g., those necessary to facilitate wide-scale roll-out) all may significantly change the average cost of programme delivery. Second, health gains are contingent on participant characteristics and engagement which likely vary across settings. Workers that benefit most from reductions in sedentary time and those receptive and motivated to maintain sedentary behaviour changes appear to be key sub-groups to consider. Thus, a targeted approach to implementing SWAL-interventions may be warranted, with those office teams with older staff, pre-existing conditions that benefit from activity, and those exposed to prolonged periods of sitting prioritised. Third, a wider perspective and further extrapolation of potential cost and HRQoL impacts from SWAL-programmes (e.g., long-term mitigation of non-fatal events, productivity, absenteeism) would give a more comprehensive account of the implications SWAL-programmes have on office-workers and their employers. Finally, cost-sharing arrangements and private procurements may distribute intervention costs more broadly, meaning sizeable public health gains may be achievable at a lower average public cost.

The cost-effectiveness of the SWAL programmes could not be demonstrated exclusively from an employer’s perspective. In contradiction with similar economic evaluations, moderate rises in absenteeism and largely inconsistent differences in worker performance, productivity and satisfaction suggested no meaningful positive changes in employer outcomes compared to usual services [47,57]. Nevertheless, employers may want to promote employee health per se and be interested in broader outcomes not measured in this analysis, including positive work environment, staff turnover, and company perception. 

## 5. Conclusions

Evidence from the SMART Work & Life trial and recent empirical findings of the all-cause mortality risks from sedentary behaviour suggests both SWAL interventions could be considered a cost-effective strategy from a public perspective for promoting the health of office workers in the UK. Study findings stress the importance of cost containment, maintaining sedentary behaviour reductions over time, and prioritisation of those individuals who stand to benefit the most from reductions in sedentary behaviour. Future research can go further by considering individual- and office-level mediators (factors which may explain the underlying mechanisms of treatment benefit), empirical measures of long-term effectiveness, sedentary behaviour dynamics, distributional cost-effectiveness (concerning the equity in the distribution of costs and effects), and modelling the competing chronic disease risks associated with sedentary behaviours.

## Figures and Tables

**Figure 1 ijerph-19-14861-f001:**
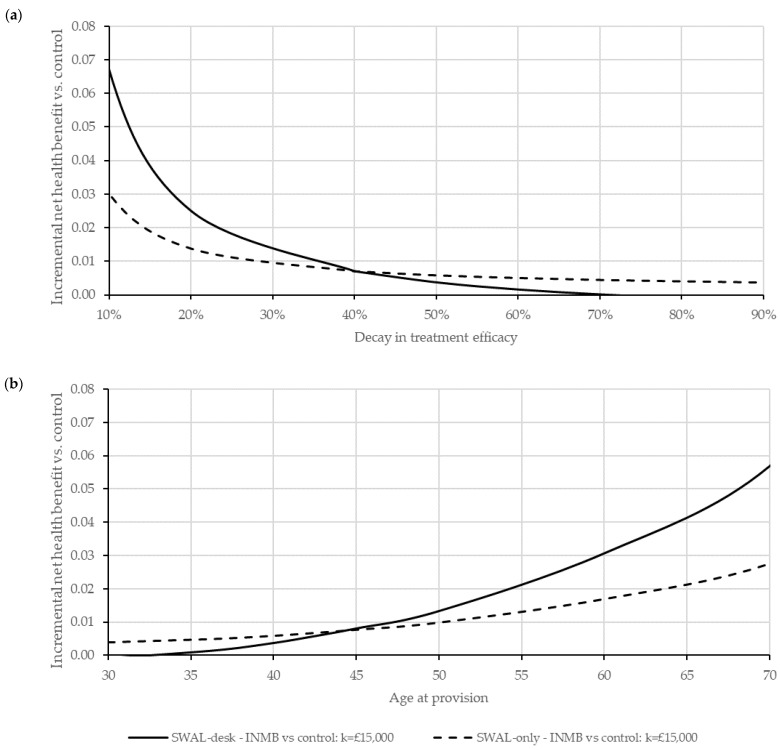
Sensitivity analyses: (**a**) Incremental net health benefits for alternative treatment decay profiles at a cost-effectiveness threshold of £15,000 per QALY; (**b**) Incremental net health benefits for alternative age profiles at a cost-effectiveness threshold of £15,000 per QALY.

**Table 1 ijerph-19-14861-t001:** Base case within-trial and lifetime cost-effectiveness results.

	Costs	QALYs	ICER	Incremental Net Health Benefit (95% CI)		
	(95% CI)	(95% CI)		k = £15,000	k = £20,000	k = £30,000
	[P (Most Costly)]	[P (Most Effective)]		[Probability of Being Cost-Effective]		
**Trial horizon**						
Control	£642.06	0.84243		-	-	-
	(505.47, 798.4)	(0.82221, 0.8642)		-	-	-
	[0.085]	[0.272]		[0.398]	[0.373]	[0.337]
SWAL-only	£691.19	0.84649	£12,090.73	0.001	0.002	0.002
	(563.3, 846.93)	(0.8265, 0.86585)		(−0.024, 0.025)	(−0.022, 0.025)	(−0.021, 0.025)
	[0.216]	[0.442]		[0.423]	[0.432]	[0.435]
SWAL-desk	£747.60	0.84187	Dominated	−0.008	−0.007	−0.007
	(641.42, 871.14)	(0.82246, 0.86006)		(−0.04, 0.023)	(−0.038, 0.024)	(−0.036, 0.024)
	[0.699]	[0.286]		[0.179]	[0.195]	[0.228]
**Life-time horizon**						
Control	£642.06	17.79359		-	-	-
	(507.14, 798.4)	(17.77337, 17.81535)		-	-	-
	[0.085]	[0.08]		[0.164]	[0.139]	[0.109]
SWAL-only	£691.19	17.80344	£4984.86	0.007	0.007	0.008
	(563.3, 846.93)	(17.78297, 17.82346)		(−0.019, 0.032)	(−0.017, 0.032)	(−0.015, 0.032)
	[0.216]	[0.336]		[0.388]	[0.376]	[0.364]
SWAL-desk	£747.60	17.80766	£13,377.90	0.007	0.009	0.011
	(641.42, 868.39)	(17.78785, 17.82522)		(−0.024, 0.041)	(−0.023, 0.042)	(−0.021, 0.043)
	[0.699]	[0.584]		[0.448]	[0.485]	[0.527]

CI: Credible interval; P: Probability; k = Cost-effectiveness threshold; ICER: Incremental cost-effectiveness ratio.

**Table 2 ijerph-19-14861-t002:** Scenario analyses.

	Costs	QALYs	ICER		Costs	QALYs	ICER
Lifetime costs				EQ-5D-5L preference values—lifetime horizon			
Control	£48,295.50	17.79359		Control	£642.06	17.82590	
SWAL	£48,361.39	17.80341	£6706	SWAL	£691.19	17.83591	£4908
SWAL-desk	£48,444.07	17.80778	£18,956	SWAL-desk	£747.60	17.84282	£8164
Linear efficacy decay (20% per annum)				EQ-5D-5L preference values—trial time horizon			
Control	£642.06	17.79359		Control	£642.06	0.87474	
SWAL	£691.19	17.80737	£3565	SWAL	£691.19	0.87896	£11,644
SWAL-desk	£747.60	17.81472	£7674	SWAL-desk	£747.60	0.87703	Dominated
70% efficacy decay (per annum)				No differential within-trial cost or QALYs estimates			
Control	£642.06	17.79359		Control	£587.42	17.79532	
SWAL	£691.19	17.80171	£6046	SWAL	£668.01	17.80111	£13,914
SWAL-desk	£747.60	17.80298	£44,597	SWAL-desk	£815.40	17.80913	£18,382
60% efficacy decay (per annum)				1.5% discount rate			
Control	£642.06	17.79359		Control	£646.26	24.20423	
SWAL	£691.19	17.80254	£5487	SWAL	£692.71	24.21901	£3143
SWAL-desk	£747.60	17.80492	£23,683	SWAL-desk	£748.50	24.22765	£6455
40% efficacy decay (per annum)				30 years-old			
Control	£642.06	17.79359		Control	£744.01	21.18629	
SWAL	£691.19	17.80520	£4231	SWAL	£783.76	21.19302	£5903
SWAL-desk	£747.60	17.81197	£8330	SWAL-desk	£826.24	21.19101	Dominated
30% efficacy decay (per annum)				40 years-old			
Control	£642.06	17.79359		Control	£667.96	18.87570	
SWAL	£691.19	17.80781	£3455	SWAL	£716.39	18.88404	£5805
SWAL-desk	£747.60	17.81906	£5016	SWAL-desk	£770.86	18.88595	£28,602
20% efficacy decay (per annum)				50 years-old			
Control	£642.06	17.79359		Control	£603.97	16.03678	
SWAL	£691.19	17.81400	£2407	SWAL	£655.99	16.05092	£3680
SWAL-desk	£747.60	17.83461	£2737	SWAL-desk	£718.06	16.05774	£9099
10% efficacy decay (per annum)				60 years-old			
Control	£642.06	17.79359		Control	£550.85	12.72221	
SWAL	£691.19	17.83321	Ext dominated	SWAL	£602.53	12.74258	£2537
SWAL-desk	£747.60	17.88543	£1080	SWAL-desk	£671.68	12.76031	£3899
0% efficacy decay (per annum)				70 years-old			
Control	£642.06	17.79359		Control	£504.50	9.05734	
SWAL	£691.19	17.99396	Ext dominated	SWAL	£560.21	9.08821	£1805
SWAL-desk	£747.60	18.26384	£209	SWAL-desk	£635.86	9.12269	£2194
Patterson et al. [19] associated all-cause mortality risks				Male			
Control	£642.06	17.79359		Control	£478.49	17.30132	
SWAL	£691.19	17.79834	£10,342	SWAL	£534.78	17.31386	£4488
SWAL-desk	£747.60	17.79494	Dominated	SWAL-desk	£613.77	17.32035	£12,186
Absenteeism cost inclusive (weighted 25% to public costs)				Female			
Control	£675.30	17.79359		Control	£726.83	17.99817	
SWAL	£726.96	17.80344	£5245	SWAL	£765.16	18.00801	£3892
SWAL-desk	£791.74	17.80766	£15,350	SWAL-desk	£810.88	18.01039	£19,280
Absenteeism cost inclusive (weighted equal to public costs)							
Control	£759.81	17.79359					
SWAL	£833.77	17.80344	£7509				
SWAL-desk	£922.12	17.80766	£20,936				

## Data Availability

The data that support the findings of this study are not openly available due to them containing information that could compromise research participant privacy/consent. Requests for participant-level quantitative data and statistical codes should be made to the corresponding author. Data requests will be put forward to members of the original trial management team who will release data on a case-by-case basis.

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
