# Peer review of "The Cost-Effectiveness of the SMART Work & Life Intervention for Reducing Sitting Time"

_ijerph, 2022, doi:10.3390/ijerph192214861_

Round 1

Reviewer 1 Report

What is the control or baseline of the experiment, ie. the reduced sitting times as a result of the  two modes of SWAL intervention compared with the baseline, such that the authors could argue for the perceived versus the resultant benefits? Without this benchmark, it is difficult to justify the significance of the research. Without a theoretical underpinning, it is hard to understand the contribution of knowledge beyond that of the cost benefit measure.  

The large team of authors for this size of survey and tracking appear unjustified. In effect it has diminished the role of the primary author if he or she existed under this arrangement, and minimised the share of each individual author. 

Author Response

Thank you for your helpful feedback.

We appreciate your comment that “control (usual practice)” is not a particularly prescriptive description of the control SWAL interventions were compared against. To address this point, we have included a more detailed description of the participants in the trial and the criteria used for recruitment (see below).

“Office groups within local councils across three areas of England were randomised to one of three arms: SWAL-desk, SWAL-only, or control. Randomisation was stratified by council area (Leicester, Liverpool, and Greater Manchester) and cluster size (<10 and ≥10 participants). Trial participants consisted of consenting, English speaking, non-pregnant and mobile (able to walk unassisted) adult office workers (≥60% full time equivalent) nested within shared office spaces [26].

Please note that we also refer to the primary trial publication which includes an in-depth view of the characteristics of those patients recruited [1].

With respect to the large number of authors, the cost-effectiveness analysis surrounding the SMART Work & Life trial was a wide-reaching multi-disciplinary effort. As the lead author, I appreciate your concern, but I am happy for my role to be amongst my 15 co-authors, without whom this work would not have been possible.

Kind regards

Reference

[1] Edwardson C L, Biddle S J H, Clemes S A, Davies M J, Dunstan D W, Eborall H et al. Effectiveness of an intervention for reducing sitting time and improving health in office workers: three arm cluster randomised controlled trial BMJ 2022; 378 :e069288 doi:10.1136/bmj-2021-069288

Reviewer 2 Report

The authors present their analysis of the implications on mortality/morbidity and the cost-effectiveness of using SWAL and SWAL desks to reduce sitting times. 

Overall, the paper is well-written, and the structure is appropriate. The study design is adequate, and the methodology seems reasonable. However, I have the following concerns:

1. The overall presentation of results could be improved by separating concerns. First, the authors should focus on assessing and quantifying the health benefits, which should be visible within the trial period. Then, once the differential improvement is established, the authors should present the cost of such improvement, including the per-group analysis. Finally, the cost-effectiveness analysis should include the cost of SWAL and SWAL desks. But those measures should be transparent. As it stands, it is tough to follow what the authors are presenting.  

2. Because of the lack of concern separation, the results in the current form seem to suggest that the author's model will always show cost-effectiveness for SWAL and SWAL desk. It is only a matter of extending the regression further into the future. So there is no new insight here. Healthier habits imply reduce in mortality and morbidities. In the same gist, the connection between SWAL and SWAL desks with reduced mortality and morbidity is too strong a conclusion based on a year worth of data, with a standard sample of the population. Therefore, I would argue that the paper would be better if the health improvements were evaluated at the end of the trial period. And only the potential cost savings for the health system is generated for the length-of-life horizon. 

Author Response

Thank you for your helpful feedback. 

Please find our responses to each of your comments below.

  1. As you have suggested, we have ordered our findings into your specified concerns: 1) outcomes (Section 3.2): what were the health effects (the within-trial changes in health-related quality of life (HRQoL), measured via the EQ5D-5L, and the changes in sitting time which are used to model longer-term health impacts); 2) resource use and costs (Section 3.3.): what additional resources and costs were expended from implementing each SWAL programme; 3) the cost-effectiveness findings (Section 3.4): what was the cost-effectiveness of each SWAL intervention.

Please note that we have tried to quantify the health benefits within trial (reported in Appendix Table 6), and those over a life-time horizon (reported in Table 1); note both per group and per ITT individual costs are reported in Appendix Table 2, and the cost of SWAL and SWAL desks are inclusive within the cost-effectiveness analysis.

  1. We apologise for any confusion regarding our modelling. The regression models are used to estimate within-trial costs and outcomes (QALYs) such that we can attain estimates that control for potential confounders and important baseline discrepancies (e.g., in baseline HRQoL) between arms. Please note that the estimated treatment-specific effects on within-trial costs and HRQoL outcomes between arms are not brought forward beyond the trial time horizon. Rather it is our decision analytic model which extrapolates the changes in sedentary behaviour associated with each intervention into QALYs (and costs in a scenario) using an established estimate of the dose response relationship between (objectively measured) sedentary behaviour and all-cause mortality [1]. We have evaluated costs and QALYs both at the end of the trial horizon (Table 1), and over a life-time horizon (Table 2). 

We hope this provides clarity. If there is any specific changes we could make to the results to help make this clearer then please let us know.

Kind regards

[1] Ekelund U, Tarp J, Steene-Johannessen J, Hansen B H, Jefferis B, Fagerland M W et al. Dose-response associations between accelerometry measured physical activity and sedentary time and all cause mortality: systematic review and harmonised meta-analysis BMJ 2019; 366 :l4570 doi:10.1136/bmj.l4570

Reviewer 3 Report

1.       The authors should explain further why the numbers of people in the three-arm cluster for the SMART Work & Life trial are different; moreover, they also should explain how the people who attended the trial be divided into the three-arm cluster.

2.       The authors should describe the qualification to be a trial participant in the SMART Work & Life trial.

3.       Line 116 mentions “Supplemental Table 1A”; however, there is no Table 1A in Appendix A.

4.       Some abbreviations do not have their corresponding full texts when they appeared first in the manuscript.

5.       About EQ-5D-5L responses, the authors should have more descriptions to explain them or have an example to demonstrate them.

6.       The authors should explain the QALY calculation in the SMART Work & Life trial further; especially, the QALY calculation relates to the generalised linear regression models.

7.       The authors should have more detailed descriptions to explain the analysis methods/models in Subsection 2.4.

8.       The authors should arrange the figures and tables shown in the supplementary file in the manuscript.

9.       Table 3 in the supplementary file is not mentioned in the manuscript. Does it require to appear in the supplementary file?

10.    The authors should explain how they set the parameter values shown in Appendix Table 5.

11.    What are the meanings of the three numbers in each box in the last three columns of Table 1?

12.    Many “SWAL-only”, “SWAL (No desk)”, and “SWAL” appear in the manuscript and the supplementary file. Do they mean the same thing? If they mean the same thing; the authors should use a unified term to express “SWAL-only”, “SWAL (No desk)”, and “SWAL” in the manuscript and the supplementary file.

13. Lines 263-264 mention “Appendix Figures 5-6”; however, no Figure 6 in the supplementary file.

14.     The “4.3 Policy Implications” in line 354 should be “4.4 Policy Implications”.

15.     Appendix Tables 6, 8, and 9 should have a legend to explain the meanings of the abbreviations.

Author Response

Thank you for your helpful feedback. Please find our responses to each of your comments below.

  1. The authors should explain further why the numbers of people in the three-arm cluster for the SMART Work & Life trial are different; moreover, they also should explain how the people who attended the trial be divided into the three-arm cluster.

The differences in the numbers of people in the three-arms are relatively small considering it was a clustered (by office) randomisation (SWAL-desk (n=240), SWAL-only (n=249) and control (n=267)). These modest differences are purely as an artefact of randomisation. To provide more clarity about the participants, the randomisation procedures and how people were divided into the three-arm cluster, the following text has been added to the manuscript (in red):

Office groups within local councils across three areas of England were randomised to one of three arms: SWAL-desk, SWAL-only, or control. Randomisation was stratified by council area (Leicester, Liverpool, and Greater Manchester) and cluster size (<10 and ≥10 participants). Trial participants consisted of consenting, English speaking, non-pregnant and mobile (able to walk unassisted) adult office workers (≥60% full time equivalent) nested within shared office spaces [26].

Further details regarding the randomisation can be found in the trial protocol and primary outcomes paper which are referenced in the paper (e.g., “Further details about SWAL and the trial are available elsewhere [29]”). 

  1. The authors should describe the qualification to be a trial participant in the SMART Work & Life trial.

See point 1.

  1. Line 116 mentions “Supplemental Table 1A”; however, there is no Table 1A in Appendix A.

All tables and figures in the supplementary appendix and main text are now consistently labelled using an “S” label (S for supplementary).

E.g.   Supplemental Table S1

  1. Some abbreviations do not have their corresponding full texts when they appeared first in the manuscript.

After reviewing the document, the only case we could identify was “UK”. As such we have amended the manuscript to write “United Kingdom” in the first instance. If you are aware of any other cases, don’t hesitate to get in touch.

  1. About EQ-5D-5L responses, the authors should have more descriptions to explain them or have an example to demonstrate them.

We have amended the text to provide a more detailed description of the EQ-5D-5L:

“Within-trial health-related quality of life (HRQoL) was measured using EQ-5D-5L responses, a descriptive instrument defined by five health dimensions (mobility; self-care; usual activities; pain/discomfort and anxiety/depression) assessed over five severity levels (no problems, slight problems, moderate problems, severe problems and extreme problems)

Please note that the citation provided includes a 150 second video explaining the EQ-5D and has a variety of resources for the readership to learn more about the EQ-5D-5L should they wish to do so.

  1. The authors should explain the QALY calculation in the SMART Work & Life trial further; especially, the QALY calculation relates to the generalised linear regression models.

To make this clear, the QALY calculations follow: (1) generate observed QALYs (Section 2.2); (2) impute missing values (2.4.3); (3) estimate QALYs using regression methods on imputed data (Section 2.4.1). Specifically, regarding the GLM model for QALYs, the specification of the model is written in detail in Section 2.4.1. What may not have been clear is the link between the QALY calculation in SMART Work & Life (i.e., generating observed QALYs in the trial, Section 2.2) and the GLM regression models (which uses these derived observations as the dependant variable). We have amended our text to make this link clearer for the readership. 

"Within-trial costs and QALYs were estimated from derived observations (see Sections 2.2-2.3) for each treatment using generalised linear regression models that controlled for age, gender, ethnicity (White vs. Other), body mass index (BMI), site area (Leicester; Salford; Liverpool) and cluster size (small <10; large ≥10)."      

  1. The authors should have more detailed descriptions to explain the analysis methods/models in Subsection 2.4.

While we appreciate that our text regarding the methods is relatively concise, we have made significant efforts to provide the readership with the key references regarding each of the methods/models employed. If there remain any specific uncertainties after considering the references for each method, we will happily provide broader descriptions where necessary.

  1. The authors should arrange the figures and tables shown in the supplementary file in the manuscript.

Amendments made so supplementary materials are ordered with the text.

  1. Table 3 in the supplementary file is not mentioned in the manuscript. Does it require to appear in the supplementary file?

The text has now been amended to refer to this table:

“Imputed within-trial EQ-5D-5L scores, mapped EQ-5D-3L values and associated QALY outcomes were broadly comparable across the trial arms (Supplementary Table S2).” (Note table numbers have been amended, see comment 8).

  1. The authors should explain how they set the parameter values shown in Appendix Table 5.

All but three model parameters were specified using data and findings from the SMART Work & Life trial. These parameters were required from the trial seeing as no other sources of evidence exists for evaluating the SWAL programmes, and that model participant characteristics should be aligned with the trial. Two parameters, cost and QALY discount rates, were determined according to NICE guidelines. This justification is provided within the text: “In line with UK guidelines, costs and QALYs were discounted at 3.5% per annum [25]” (p2, line 83). The only model parameter not specified according to UK guidelines or from the trial itself was the sedentary associated all-cause hazards. The source used (Ekelund et al [1]) was discussed extensively with the authors and determined to be the best available source of evidence for the sedentary-specific objectively measured (accelerometer) dose-response relationship between sedentary time and all-cause mortality [1]. Note an alternative source was explored in scenario analysis. Thank you for bringing this to our attention, we hope this provides adequate clarity.

  1. What are the meanings of the three numbers in each box in the last three columns of Table 1?

“Net health benefit (NHB) is a summary statistic that represents the impact on population health of introducing a new intervention.  Net health benefit assumes that ‘lost health’ can be estimated as an ‘opportunity cost’ to represent the health that is foregone elsewhere as a result of moving funding to pay for a new intervention.  NHB is usually measured using QALYs and is calculated by: incremental gain in QALYs – (incremental cost / opportunity cost threshold).  A positive NHB implies that overall population health would be increased as a result of the new intervention, whilst a negative NHB implies that the health benefits of the new intervention are not sufficient to outweigh the health losses that arise from the healthcare that ceases to be funded in order to fund the new treatment.” Source: https://yhec.co.uk/glossary/net-health-benefit/

The last three columns of Table 1 report the incremental net-health benefits of SWAL-only and SWAL-desk and provides the level of uncertainty associated with each (95% credible intervals). Positive values indicate a net-health gain, negative values a health loss from diverting healthcare funding to its provision. INHB is increasing used in economic evaluation.

I hope this clears up your question.

  1. Many “SWAL-only”, “SWAL (No desk)”, and “SWAL” appear in the manuscript and the supplementary file. Do they mean the same thing? If they mean the same thing; the authors should use a unified term to express “SWAL-only”, “SWAL (No desk)”, and “SWAL” in the manuscript and the supplementary file.

We really appreciate this feedback. SWAL-only now used throughout the manuscript.

  1. Lines 263-264 mention “Appendix Figures 5-6”; however, no Figure 6 in the supplementary file.

Agreed. Amended:

Sensitivity analyses surrounding treatment efficacy decay rates and worker ages at thresholds £20,000 and £30,000 per QALY are reported in Supplementary Appendix Figures S4-S5

  1. The “4.3 Policy Implications” in line 354 should be “4.4 Policy Implications”.

Agree. Amended:

4.4 Policy Implications”

  1. Appendix Tables 6, 8, and 9 should have a legend to explain the meanings of the abbreviations.

Agreed. Abbreviations added to each as a footnote:

References:

[1] Ekelund U, Tarp J, Steene-Johannessen J, Hansen B H, Jefferis B, Fagerland M W et al. Dose-response associations between accelerometry measured physical activity and sedentary time and all cause mortality: systematic review and harmonised meta-analysis BMJ 2019; 366 :l4570 doi:10.1136/bmj.l4570

Reviewer 4 Report

The Literature Review is extensive and has very current articles. However, there are some articles that are a few years old, which may question their timeliness. For example, articles from 2004, 2005 and 2007. Is it possible for the authors to find more recent articles on the same topics?

The conclusions are very vague and should be reviewed and strengthened by the authors. It would be better if the authors put here some of the most important values obtained with this research and reinforce in this way the importance and contribution of this research to the society and scientific community.

It is recommended that authors include recommendations for future work at the end of the article. It would be very interesting if the community could continue this research and bring more added value to the scientific community and to society.

Author Response

Thank you for your helpful feedback. Please find our responses to each of your comments.

Literature recency

We have reviewed the years in question (2004, 2005, and 2007) and while these articles (see below) could be considered less contemporary, the majority are in reference to key methodological findings (Manca et al (2005); Barber et al (2004) and Ruben (2004)) whose insights are still very much applicable to our works today.

The only non-methodological work published in the years in question is by Bruggen et al (2007). Please note that while this is an older cost-effectiveness analysis, economic evaluations in sedentary-specific lifestyle interventions tend to be quite rare [1], and as such we undertook special efforts to consider a broader history of articles in the field. Also note that this article was considered for its structural assumptions (specifically, the longer-term efficacy of a life-style intervention) rather than evaluating the study and its findings more broadly.   

Thank you for bringing this to our attention, we hope this provides adequate clarity.

Article references (2004, 2005, 2007):

2007: Jacobs-Van Der Bruggen MAM, Gri¨et M, Bos G, Bemelmans WJ, Hoogenveen RT, Vijgen SM, et al. Lifestyle Interventions Are Cost-Effective in People With Different Levels of Diabetes Risk Results from a modeling study. Diabetes Care [Internet]. 2007 [cited 2021 Mar 8];30:128–34. Available from: http://care.diabetesjournals.

2005: Manca A, Hawkins N, Sculpher MJ. Estimating mean QALYs in trial-based cost-effectiveness analysis: The importance of controlling for baseline utility. Health Econ [Internet]. 2005 May [cited 2021 Mar 2];14(5):487–96. Available from: https://pubmed.ncbi.nlm.nih.gov/15497198/

2004: Barber J, Thompson S. Multiple regression of cost data: Use of generalised linear models. J Heal Serv Res Policy. 2004 Oct;9(4):197–204.

2004: Rubin D. Multiple imputation for nonresponse in surveys. 2004.

Future research and study conclusions

We appreciate your feedback regarding the potential vagueness of our findings. Given that the cost-effectiveness conclusions are predicated on trial-specific costings and non-empirical assumptions regarding the longer-term effectiveness of the intervention on participants, we believe a degree of cautionary language is necessary. Where possible however we have tried to reword our findings into more assertive forms of language.

As requested, recommendations for future work have been included at the end of the article.

Future research can go further by considering individual- and office-level mediators (factors which may explain the underlying mechanisms of treatment benefit), empirical measures of long-term effectiveness, sedentary behaviour dynamics, distributional cost-effectiveness (concerning the equity in the distribution of costs and effects), and modelling the competing chronic disease risks associated with sedentary behaviours.

References:

[1] Nguyen P, Le LK, Ananthapavan J, Gao L, Dunstan DW, Moodie M. Economics of sedentary behaviour: A systematic review of cost of illness, cost-effectiveness, and return on investment studies. Preventive Medicine. 2022 Mar 1;156:106964.

Round 2

Reviewer 1 Report

The authors are requested to provide details of the SWAL desk and SWAL only with reference to the resultant time saving of standing over sitting, i.e. how to explain the health benefits of SWAL_only and SWAL_desk by time saving (22.2 minutes versus 63.7 minutes). How could this piece of analysis weigh against the cost-benefit analysis in the determination of SWAL_only over SWAL_desk in the Discussion part of the manuscript.

Author Response

Thank you for your thorough and well considered feedback. As requested, we have made the following amendments to the manuscript:

(1): Specific details are now provided regarding the health benefits associated with the reductions in sedentary time from SWAL-only and SWAL-desk:

“At model baseline (12-month follow-up), reductions in sitting time for SWAL-desk and SWAL-only translated into 23.8% and 10.5% reductions in the relative-risk of all-cause mortality compared to control, respectively.” (page 5, Section 3.2 Outcomes)

(2): The discussion now includes a comparison of the cost-effectiveness between SWAL-only and SWAL-desk:

“Compared to SWAL-only, SWAL-desk could be considered cost-effective from the public perspective provided desk-related costs fall below £202 and those additional reductions in sedentary time associated with its application are adequately maintained (≥39% per annum).” (page 9, Section 4.1 Key findings)

Kind regards.

Reviewer 3 Report

The authors depend on all comments to revise the manuscript.

Author Response

Thank you for your thorough and well considered feedback. We have revised the manuscript in accordance with all comments we have received from yourself and other peer reviewers. We found no inconsistencies between reviewer comments and believe the manuscript is now markedly improved as a result of the inputs we have received. If you would like to raise anything further, please don’t hesitate to get in touch.

Kind regards.